# Effect of the Rearing Managements Applied during Heifers’ Whole Life on Quality Traits of Five Muscles of the Beef Rib

**DOI:** 10.3390/foods8050157

**Published:** 2019-05-10

**Authors:** Julien Soulat, Valérie Monteils, Brigitte Picard

**Affiliations:** Université Clermont Auvergne, INRA, VetAgro Sup, UMR Herbivores, F-63122 Saint-Genès-Champanelle, France; julien.soulat@inra.fr (J.S.); valerie.monteils@vetagro-sup.fr (V.M.)

**Keywords:** rearing managements, chuck sale section, meat sensory properties, meat rheological properties, color attributes, longissimus muscle, complexus muscle, infraspinatus muscle, rhomboideus muscle, serratus ventralis muscle

## Abstract

The aim of this work was to study the effects of four different rearing managements applied during the heifers’ whole life period (WLP) on muscles from ribs in the chuck sale section. The characteristics of meat studied were the sensory, rheological, and color of the longissimus muscle (LM) and the rheological traits of four other muscles: complexus, infraspinatus, rhomboideus, and serratus ventralis. The main results showed that WLP rearing managements did not significantly impact the tenderness (sensory or rheological analyses) of the rib muscles. The LM had high (*p* ≤ 0.05) typical flavor and was appreciated when heifers received a WLP rearing management characterized by a short pasture duration during the heifers’ whole life (WLP-E). The heifers’ management characterized by a long pasture duration during their life (WLP-A) or by a diet composed mainly of hay during the growth and fattening periods (WLP-F), had lower typical flavor and were less appreciated than those with WLP-E management. Moreover, the LM color was redder for heifers of WLP-E than those of the WLP-A and WLP-F groups. This study confirmed that it is possible to obtain similar meat qualities with different rearing managements.

## 1. Introduction

Beef carcasses are composed of many skeletal muscles [1] with different properties, e.g., structural, metabolic and contractile [2,3]. After beef carcass cuts, the wholesale cuts purchased by consumers could be composed of different muscles, e.g., ribs and short ribs of beef. It is well known that meat quality traits can be impacted by different factors, e.g., animal type (sex and breed) [4,5,6], stress (transport, slaughter condition) [7,8,9] and rearing managements [10,11,12]. Many studies showed that different rearing factors observed during the fattening period (e.g., slaughter age [6,13], fattening duration [14,15], and fattening diet [16,17,18]) had an effect on meat quality traits. Recent studies have shown that the rearing management (combining different rearing factors) applied during the animal’s whole life period (i.e., from birth to slaughter, whole life period (WLP)) could have an impact on the carcass and/or meat traits [10,11,12]. These recent studies had observed the effect of the WLP rearing managements on the flank steak (rectus abdominis muscle, RA) [10,11]. The aim of the present work was to study the effects of the WLP rearing managements defined by Soulat et al. [10] on the longissimus muscle (LM) traits (sensory, rheological and color properties), in the ribs of the chuck sale section. Moreover, the effects of the WLP rearing managements were also studied on the toughness (rheological) of four other muscles composing these ribs: complexus, infraspinatus, rhomboideus, and serratus ventralis. In the literature, the LM is considered as a reference muscle and the other muscles composing the rib have been poorly studied. One originality of this work was to observe the effects of the same WLP rearing management on these different muscles.

## 2. Materials and Methods

### 2.1. Animals and Rearing Managements

The present study was realized from the same experiment presented by Soulat et al. [10] in partnership with the protected geographical indication (PGI) Fleur d’Aubrac. Briefly, this PGI produces exclusively crossbreed Charolais with Aubrac heifers for the meat production. The breeding of the heifers was realized respecting the design brief of this PGI, which is based mainly on a grass diet (conserved and pasture) [19]. The heifers (*n* = 48) considered in this work, were produced in eight commercial farms. For this study, the same four rearing managements described by Soulat et al. [10] applied during the animal’s whole life period were considered: WLP-A, WLP-D, WLP-E and WLP-F (Figure 1). Briefly, these WLP were the combination of different rearing management clusters characterizing three periods of the heifers’ life: pre-weaning, growth and fattening [10]. The clusters at each period of heifers’ life were obtained statistically from the data collected during the surveys. The heifers receiving the rearing management WLP-A and WLP-D had the same pre-weaning management (PWP-clust1). The rearing managements applied during the growth and the fattening periods (GP-clust1 and FP-clust3) were specific to the WLP-A rearing management. The WLP-D had specific management during the growth period of the heifers (GP-clust3). The WLP-D and WLP-F had the same management during the fattening period (FP-clust2). The WLP-E and WLP-F rearing managements had the same rearing management during the pre-weaning period (PWP-clus2) and the growth period (GP-clust2). The rearing management applied during the fattening period of the heifers (FP-clust1) was specific to the WLP-E rearing management. Briefly, the WLP-A and WLP-E rearing managements were mainly characterized by a long and a short pasture duration during the heifers’ whole life, respectively. The WLP-D rearing management was mainly characterized by a high concentrate quantity intake by the heifers during the growth and the fattening periods. The WLP-F rearing management was mainly characterized by a diet composed mainly of hay during the growth and the fattening periods. 

### 2.2. Animals Slaughtering, Muscle Sampling and Meat Quality Evaluation

The heifers were slaughtered by exsanguination after stunning in the same industrial slaughterhouse (Abattoir du Gévaudan, Antrenas, France) as described in Soulat et al. [10]. 

Twenty-four hours post-mortem, two beef ribs (the 5th and 4th ribs), localized in the chuck sale section, were collected from the right-hand side of the carcass. Two hours after excision from the carcass, the color was measured on the LM of the rib (side of the cutting section between the 6th and the 5th ribs). The color was expressed in CIE L*a*b* units [20], using a spectrophotometer (Konica Minolta CM-600d, Osaka, Japan) (light source D65, 8 mm diameter measurement area, and 0° standard observer). Before each measurement session, the spectrophotometer was calibrated by performing a black and a white calibration. Five measurements (randomly distributed on the muscle) per LM were realized to characterize the color of this muscle. The chroma (C*) and hue angle (*h**) were calculated from the a* and b* values, as realized by Gagaoua et al. [21].

The rib samples of each animal were vacuum-packaged and chilled at 4 °C during 14 days for aging. At the end of the aging period, the samples were frozen at −20 °C until the analyses [22].

In this study, the sensory evaluation was only realized on the LM by a trained tasting panel (15 members), using a monadic test. The sensory evaluation was realized using the same process described by Soulat et al. [10]. Briefly, the members of the trained tasting panel had 20 training sessions (between 1 and 1.5 h per session) before starting the sensory evaluation of the LM, in accordance with ISO 8586 [23]. Before each tasting session, the ribs were thawed and dissected to separate the different muscles. The LM samples were cut into two sub-samples: The first for the sensory evaluation, and the second for the shear force measurement. For sensory evaluation, the LM samples were cut into steaks, and cooked on a double-face grill to reach an internal temperature of 55 °C. Then, samples were cut (size 15 × 20 × 20 mm) and 3 or 4 pieces were served to each member of the trained panel. At each tasting session, a Latin square presentation was used to evaluate the sensory traits of five samples. 

The trained panel evaluated six sensory descriptors: The initial tenderness, overall tenderness, initial juiciness, overall juiciness, typical flavor, and overall acceptability. The initial tenderness and juiciness were defined as an evaluation at the first bite of the tenderness, and juiciness, respectively. In contrast, the overall tenderness and juiciness were an evaluation of the tenderness and the juiciness, respectively, before swallowing the meat sample. These six sensory descriptors were measured on a 10-point non-graduated scale from a score of 0 (tough, dry, slight, and highly disliked) to a score of 10 (very tender, very juicy, strong, and highly liked).

The shear force was measured on five muscles of the rib, located in the chuck sale section: The complexus (CP), infraspinatus (IF), longissimus (LM), rhomboideus (RH), and serratus ventralis (SV) (Figure 2). The shear force was evaluated using a Warner–Braztler apparatus (EZ-SX set assay EU RoHS, Shimadzu, Kyoto, Japan) on raw meat [24]. For each muscle from the rib, at least five meat portions (length: 1.5 to 3 cm, width: 1 cm and thickness: 0.5 to 1 cm) were cut perpendicular to the fibers [25]. From the different measurements per muscle, the shear force was calculated using the Trapezium X 1-5.1 software (Shimadzu, Kyoto, Japan).

The distribution of the meat rib traits considered in this study are presented in Table 1.

### 2.3. Statistical Analyses

The statistical analyses were realized using R 3.5.2 software (R Core Team, Vienna, Austria) [26].

A descriptive analysis of the dataset, using quantile-quantile plots, was performed to observe the normality of the distribution [27]. Then, for each meat trait, an analysis of variance (ANOVA) was performed to evaluate their dependence on the four considered WLP rearing managements. In all ANOVA, the farm effect was tested. If it was significant, it was considered as a random effect in the ANOVA. In the ANOVAs of the sensory parameters, the effect of the member of the trained tasting panel and the animal effect were considered as random effects. For the ANOVAs without a random effect, if the result was significant a post-hoc Tukey test was performed, using the agricolae package [28]. The ANOVAs containing random effects were developed using the lmerTest package [29]. If the results of these ANOVAs were significant, a post-hoc Tukey test was also performed, using the multcomp package [30].

An effect was considered significant at *p* ≤ 0.05 and a tendency was considered for 0.05 < *p* ≤ 0.10.

## 3. Results and Discussion

The four WLP rearing managements considered, in this work, had no significant effect on the tenderness of the LM, as measured by a trained tasting panel or by shear force (Table 2). This result is in accordance with those of Soulat et al. [10] who observed no significant effect on the tenderness of the flank steak (RA muscle). However, tendencies (*p* ≤ 0.10) were displayed for the initial tenderness and the shear force. The heifers receiving the WLP-E or WLP-F rearing managements (Figure 1) tended (*p* ≤ 0.10) to produce LM with a higher initial tenderness than those receiving the WLP-A and WLP-D rearing managements (Table 2). The WLP-E and WLP-F rearing managements were characterized by a short pasture duration during the heifers’ whole life and by a diet composed mainly of hay during the growth and fattening periods, respectively [10]. However, the WLP-A and WLP-F rearing managements were characterized by a long pasture duration during the heifers’ whole life and by a high concentrate quantity intake during the growth and fattening periods. The raw LM also tended to be less tough when the heifers received the WLP-E or WLP-F rearing managements (Table 2). The importance of collagen in meat could explain this weak difference. However, this tendency was not found for the overall tenderness, as evaluated by the trained tasting panel. It is possible that the cooking mitigated this tendency. The four considered rearing managements did not also significantly (*p* > 0.05) affect the toughness (evaluated by shear force) of the other rib muscles uncooked: CP, IF, RH, and SV. Based on our overall results, the variation of rearing managements applied at the different key periods of the heifers’ life seems to have no impact on the toughness of the five raw rib muscles. Our results show that it is possible to obtain the same overall tenderness of the LM with different rearing managements. The muscle traits could have more impact on the tenderness than the rearing management applied during the heifers’ whole life.

The LM had a significant (*p* < 0.05) higher initial juiciness for heifers receiving the WLP-E rearing management than those receiving the WLP-A (Table 2). Nevertheless, the four WLP rearing managements had no significant effect on the overall juiciness. These results are in accordance with those of Soulat et al. [10] for the flank steak.

According to our results, the WLP-E rearing management produced an LM with a significantly (*p* < 0.05) higher typical flavor and higher appreciated meat than the WLP-A and WLP-F rearing managements (Table 2). The WLP-E and WLP-F rearing managements differ only by the rearing management applied during the fattening period, which was different (Figure 1). The fattening period duration was significantly (*p* < 0.05) longer in the WLP-E rearing management than the WLP-F and the WLP-A managements [10]. In cull cows, studies of the literature showed a significant increase in the LM flavor intensity when the fattening duration was longer [15,31]. Moreover, in the flavor prediction model developed by Soulat et al. [32], an increase of the fattening duration allowed to increase the LM flavor intensity in the cull cows. The main forages in the WLP-E and WLP-F rearing managements were different: Hay or hay and wrapped haylage, respectively. According to the results of different studies, the composition of the fattening diet could have no impact (*p* > 0.05) on the LM flavor intensity, in heifers and steers [33,34,35]. During the fattening period, the heifers from the WLP-F rearing management pastured, whereas those from the WLP-E were inside [10]. With less walking during the fattening period for the heifers from the WLP-E rearing management, it was possible that their LM was less oxidative and had more intramuscular lipid content than those from the WLP-F. In their study, Jurie et al. [36] showed that the RA muscle was more oxidative (an increase of isocitrate dehydrogenase concentration) when the steers moved during pasture. Moreover, studies showed that cattle with fattening management with pasture produced leaner carcasses [37,38]. In consequence, these carcasses could have less marbling. However, Soulat et al. [10] did not find a significant effect (*p* > 0.05) of the four WLP rearing managements on the flavor intensity of flank steak. The WLP-A and WLP-E rearing managements had no rearing managements in common during the different key periods of the heifers’ life (Figure 1). During the pre-weaning period, the heifers from the WLP-A rearing management had an average daily gain (ADG) lower than those from the WLP-E [10]. According to the results of Hennessy et al. [39], the meat flavor intensity was lower when cattle had a quick growth before weaning. Our results are not in accordance with these results. This difference could be explained by the fact that the heifers of our study did not have the same rearing management during the growth and the fattening periods. During the fattening period, the heifers from the WLP-A rearing management ingested a lower concentrate quantity than those from the WLP-E rearing management. Different studies showed that an increase of the concentrate quantity in the fattening diet increased the LM flavor intensity in steers [16,40]. However, other studies did not observe an effect of the concentrate quantity in the fattening diet on the LM flavor intensity [18,41,42]. As the WLP-F rearing management, the fattening period duration was shorter in the WLP-A than in the WLP-E rearing management [10]. The main fattening managements realized in the WLP-A was pasture or pasture and housing, whereas, it was housing in the WLP-E (Figure 1). In their study, Duckett et al. [43] observed that the LM flavor intensity was significantly higher when the fattening management was a concentrate ration compared to mixed pasture. In the WLP-E rearing management, the heifers were slaughtered heavier than those from the WLP-A. According to the results of different studies, the slaughter weight had no impact on the LM flavor intensity, in young bulls and steers [44,45]. In the study of Oury et al. [46] and Soulat et al. [10], in heifers, the flavor intensity of flank steak was not impacted (*p* > 0.05) by the rearing management applied during the heifers’ whole life. Concerning the flavor, the LM seems to be more sensitive to the WLP rearing management variations compared to the flank steak. 

In our study, the effect of the WLP rearing managements was the same on the overall acceptability and the typical flavor of the LM (Table 2). As the WLP rearing managements did not affect the tenderness and the juiciness of the LM, we suppose that the overall acceptability of the LM was strongly linked with the typical flavor. According to the results of different studies in cattle, the slaughter weight [45], the fattening diet composition or the fattening management [33,34,47] did not impact the LM overall acceptability. 

For the LM color, the four rearing managements had no significant (*p* > 0.05) effect on the L*, b*, and *h** color parameters (Table 2). According to the a* values, the LM was significantly redder for heifers receiving the WLP-E than those receiving the WLP-A and WLP-D rearing managements. The heifers receiving the WLP-E management also produced LM with greater red color intensity, according to the C* values, than those receiving the WLP-A. To our knowledge, the impact of the pre-weaning and growth period on the LM color have not been studied. However, many studies showed that the fattening duration [48] and the composition of the fattening diet [18,49,50] did not affect (*p* > 0.05) the LM color. However, many studies showed that fattening management had a significant (*p* < 0.05) effect on the LM color [37,43,51]. In these studies, the cattle with a pasture period during their fattening produced LM meat with a lower a* than those fattened in housing. The L* and b* parameters were also impacted by the rearing managements applied during the fattening period. In our study, the heifers from the WLP-E rearing management had higher a* value than those receiving the WLP-A and WLP-F rearing managements (Table 2). This result is in accordance with the literature. In the WLP-E rearing management, the heifers were not pastured during their fattening compared to the heifers receiving the WLP-A and WLP-F rearing managements (Figure 1). In our study, in contrast to the results of Duckett et al. [43], Cozzi et al. [51], and Huuskonen et al. [37], the L* and b* color parameters were not impacted by the WLP rearing managements. As rearing management is multifactorial, it is possible that the effect of the fattening management was mitigated by this combination with the other rearing factors. 

Considering the WLP rearing managements, it is difficult to identify the rearing factors, which had the greatest impact on meat quality. In our study, the rearing factors varied independently, whereas, in the literature, many studies observed the effect of one or two rearing factors in an experimental condition. It is possible that some rearing factors have antagonist effects on meat quality during the life of the heifers. The difficulty of interpretation of our results was considering all the combinations of all the rearing factors. Moreover, the impact of rearing management applied during a period of the heifers’ life can be mitigated or amplified by the rearing management applied during another period of life.

## 4. Conclusions

The originality of this study was to observe the impact of four different rearing managements applied during the heifers’ whole life on the traits of the LM (sensory, rheological, and color) in the ribs of the chuck sale section. In accordance with the results of Soulat et al. [10], these new results showed that different rearing managements applied during the heifers’ whole life obtained the same meat quality, particularly, the tenderness and juiciness. Finally, all the results of this experiment showed that the LM traits seem to be more sensitive to variations of the rearing managements than the flank steak, in particular, for the typical flavor, the overall acceptability, and the a* value. Our results also showed that these rearing managements did not significantly impact the toughness of the other four rib muscles uncooked: complexus, infraspinatus, rhomboideus, and serratus ventralis. These results demonstrate that farmers could adapt, to a certain extent, their rearing managements during the heifers’ whole life, according to the hazards (e.g., drought, price of concentrates), with limited consequences on the meat quality. 

## Figures and Tables

**Figure 1 foods-08-00157-f001:**
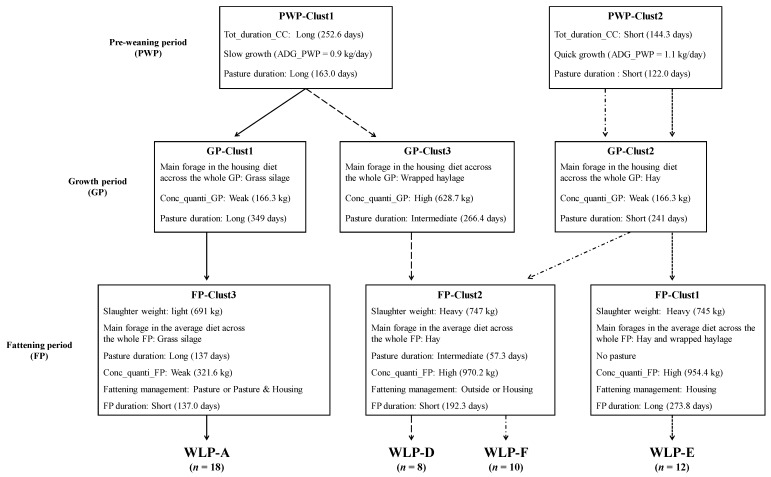
Description of the four rearing managements applied during the heifers’ whole life period (WLP) defined by Soulat et al. [10] with a focus on the rearing factors with the most differences between groups. Tot_duration_CC: Total time spent by the calf with her mother between the birth and the weaning, ADG: Average daily gain, Conc_quanti: Total concentrate quantity intake during the period of heifers’ life.

**Figure 2 foods-08-00157-f002:**
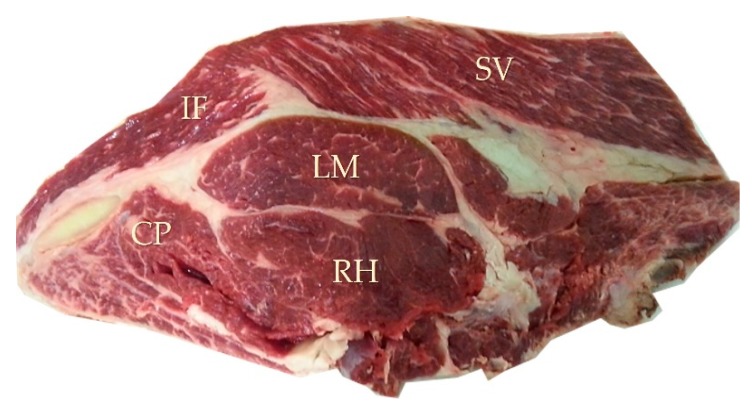
Localization of five rib muscles: complexus (CP), infraspinatus (IF), longissimus (LM), rhomboïdeus (RH), and serratus ventralis (SV).

**Table 1 foods-08-00157-t001:** Description of the rib muscle traits.

Meat Traits	Mean	SD	Min	Max
Sensory description of longissimus muscle (scale 0–10) ^1^				
Initial tenderness	7.25	1.44	1.42	10.00
Overall tenderness	7.10	1.62	1.45	10.00
Initial juiciness	6.63	1.54	1.23	9.95
Overall juiciness	6.65	1.64	0.61	9.97
Typical flavor	6.60	1.57	0.84	9.99
Overall acceptability	6.37	1.68	0.27	9.96
Color of longissimus muscle				
L*	32.90	2.80	27.70	41.32
a*	18.23	2.53	12.67	26.52
b*	17.82	2.47	8.64	21.39
C*	25.10	4.39	3.06	33.03
*h**	43.50	6.93	3.98	50.42
Shear force (N/cm)				
Complexus	61.96	13.67	30.93	87.62
Infraspinatus	99.45	45.61	45.49	278.23
Longissimus	45.92	13.38	24.03	88.54
Rhomboideus	61.20	16.95	13.32	112.93
Serratus ventralis	56.23	18.52	31.18	125.29

SD: Standard deviation. Min: Minimum. Max: Maximum. ^1^, Scale for initial tenderness, overall tenderness, initial juiciness, overall juiciness, typical flavor, overall acceptability: 0 = tough, dry, slight, and highly disliked and 10 = very tender, very juicy, strong, and highly liked.

**Table 2 foods-08-00157-t002:** Impact of the four rearing managements applied during the heifers’ whole life period (WLP) on the traits of five rib muscles.

Meat Traits				WLP Rearing Managements ^1^									*p*
	WLP-A (*n* = 18)			WLP-D (*n* = 8)			WLP-E (*n* = 12)			WLP-F (*n* = 10)			
	Mean	SD	SE	Mean	SD	SE	Mean	SD	SE	Mean	SD	SE	
Sensory description of longissimus muscle (scale 0–10) ^2^													
Initial tenderness	7.10	1.55	0.24	7.04	1.43	0.10	7.53	1.30	0.10	7.34	1.34	0.11	0.06
Overall tenderness	6.91	1.74	0.26	6.98	1.51	0.11	7.36	1.47	0.11	7.22	1.61	0.13	0.21
Initial juiciness	6.46 ^a^	1.58	0.25	6.61 ^ab^	1.52	0.11	6.98 ^b^	1.41	0.11	6.52 ^ab^	1.59	0.13	0.04
Overall juiciness	6.53	1.75	0.28	6.53	1.51	0.11	6.97	1.50	0.11	6.58	1.67	0.14	0.12
Typical flavor	6.52 ^a^	1.60	0.25	6.55 ^ab^	1.32	0.09	7.00 ^b^	1.47	0.11	6.32 ^a^	1.75	0.14	0.01
Overall acceptability	6.14 ^a^	1.68	0.26	6.29 ^ab^	1.49	0.11	6.92 ^b^	1.67	0.13	6.24 ^a^	1.71	0.14	0.005
Color of longissimus muscle													
L*	32.36	2.54	0.60	33.38	2.75	0.97	32.77	3.70	1.07	33.65	2.16	0.68	0.66
a*	17.07 ^a^	2.02	0.48	18.67 ^ab^	1.67	0.59	20.51 ^b^	2.78	0.80	17.23 ^a^	1.79	0.57	<0.001
b*	16.90	3.31	0.78	18.73	1.29	0.46	18.55	1.89	0.54	17.84	1.59	0.50	0.21
C*	24.10 ^a^	3.32	0.78	26.45 ^ab^	1.99	0.70	27.70 ^b^	2.93	0.85	24.84 ^ab^	1.99	0.63	0.008
*h**	44.36	5.11	1.20	45.13	1.53	0.54	42.24	3.40	0.98	46.03	3.06	0.98	0.14
Shear force (N/cm)													
Complexus muscle	62.21	13.01	3.07	61.27	18.12	6.41	64.17	13.85	4.00	59.41	12.33	3.90	0.88
Infraspinatus muscle	115.92	48.42	11.41	97.00	69.56	24.59	80.64	21.54	6.22	94.33	32.50	10.28	0.21
Longissimus muscle	50.17	15.47	3.65	51.27	17.67	6.25	40.24	5.75	1.66	40.81	8.09	2.56	0.08
Rhomboideus muscle	61.74	13.12	3.09	64.38	28.49	10.07	58.90	14.16	4.09	60.45	16.82	5.32	0.92
Serratus ventralis muscle	59.60	15.95	3.76	46.01	16.20	5.73	62.34	23.29	6.72	51.00	15.74	4.98	0.16

*n*: number of heifers. SD: Standard deviation. SE: Standard error. Values followed by different letters (a, b) are significantly different from each other at *p* ≤ 0.05. ^1^, The WLP rearing managements were the rearing managements described by Soulat et al. [10]. ^2^, Scale for initial tenderness, overall tenderness, initial juiciness, overall juiciness, typical flavor, overall acceptability: 0 = tough, dry, slight, and highly disliked and 10 = very tender, very juicy, strong, and highly liked.

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
