# Peer review of "Effect of the Rearing Managements Applied during Heifers’ Whole Life on Quality Traits of Five Muscles of the Beef Rib"

_foods, 2019, doi:10.3390/foods8050157_

Round 1
Reviewer 1 Report
General comments
1. The authors need to reconsider the statistical analysis they conducted on trained panel meat traits. There is no way the same means can have different superscripts. More than that, the treatment effects on initial juiciness, typical flavor, and overall acceptability are not realistic with the given SEM and P value.
2. How and why did you choose these Rearing managements (treatments)? On what basis did you choose the clusters? What is the meaning of your treatment arrangement in an industry point of view?
3. Explain your experimental design (CRD? RCB?) Was there any blocking in this study? Why did the number of animals differ hugely across treatments? Did you do a power analysis to determine the sample size before you started the experiment? If yes, please explain.
4. Authors need to provide P values for each comparison you make in the result section.
5. There are spelling and grammatic errors throughout the manuscript, some of which have been pointed out in the specific comments.
Specific comments
Line 2-4: Title: You need to mention ‘sensory, rheological, and color attributes’ since you focused on those.
Line 11-12: The studied traits were the sensory, rheological and color traits of longissimus muscle (LM) and the rheological traits of four other muscles….
‘Traits’ repeated. May need to restructure
Coma needed after rheological
Line 13: complexus, infraspinatus, longissimus, rhomboideus and serratus….
Coma needed after rhomboideus
This punctuation mistake was observed throughout the manuscript. Please correct all of them.
Line 32-33: Many studies of the literature……
Many studies showed….
Line 35: Recent studies had shown…..
Recent studies have….
Line 40: sensory, rheological and color proprieties
Wrong spelling
Line 42: four others muscles
four other muscles
Line 50-51: Farmers respect a specific design brief [19] based mainly on a grass diet (conserved and pasture).
What is specific design brief ? What does this sentence mean?
Line 61: The WLP-E and WLP-F rearing managements had the rearing management….
The WLP-E and WLP-F rearing managements had the SAME rearing management….
Line
Lines 89-90: Briefly, the members of the trained tasting panel had training sessions before
90 starting the sensory evaluation of the LM.
Please explain the training. How many sessions? How many hours? What were the references used?
Lines 92-93: For sensory evaluation, the LM samples were cut into steaks and cooked on a double-face grill to reach an internal temperature of 55°C.
What was the serving size? A whole steak? Or cubes? If yes what size?
Line 135: of the LM eather…
Change this.
Author Response
Reviewer 1 :
Open Review
(x) I would not like to sign my review report
( ) I would like to sign my review report
English language and style
( ) Extensive editing of English language and style required
(x) Moderate English changes required
( ) English language and style are fine/minor spell check required
( ) I don't feel qualified to judge about the English language and style
Yes | Can be improved | Must be improved | Not applicable | |
Does the introduction provide sufficient background and include all relevant references? | ( ) | ( ) | (x) | ( ) |
Is the research design appropriate? | ( ) | ( ) | (x) | ( ) |
Are the methods adequately described? | ( ) | ( ) | (x) | ( ) |
Are the results clearly presented? | ( ) | ( ) | (x) | ( ) |
Are the conclusions supported by the results? | ( ) | ( ) | ( ) | (x) |
Comments and Suggestions for Authors
General comments
1. The authors need to reconsider the statistical analysis they conducted on trained panel meat traits. There is no way the same means can have different superscripts. More than that, the treatment effects on initial juiciness, typical flavor, and overall acceptability are not realistic with the given SEM and P value.
We have checked all data in our tables. We added a decimal to our numbers to show why we found significant differences between the means. We recalculated all SEM and P-values in order to verify our results which we validate.
2. How and why did you choose these Rearing managements (treatments)?
The rearing managements chosen in the present article are those described in Soulat et al. (2018) as specified in lines 38-41. In the article of Soulat et al, 2018 in Foods a statistical analysis was applied to create clusters from rearing factors applied during the whole life of heifers and look at the differences in the flank steak qualities (rectus abdominis muscle) between these clusters. In the present article, in complement, we looked at the differences between the same clusters on the qualities of longissimus muscle and four other muscles of the Beef Rib. The text has been modified to better explain this approach. We added information in the Materials and Methods section (lines 56 – 58).
Soulat, J., B. Picard, S. Léger, M.-P. Ellies-Oury, and V. Monteils. 2018. Preliminary Study to Determinate the Effect of the Rearing Managements Applied during Heifers’ Whole Life on Carcass and Flank Steak Quality. Foods 7:160.
On what basis did you choose the clusters?
As described in Soulat et al. (2018) during this experiment, we have realized surveys to collect information (qualitative and quantitative) on the rearing management applied at different periods of the heifers’ whole life. Three periods were considered: pre-weaning, growth and fattening. The clusters were obtained statistically for each life period, using a factor analysis for mixed data followed by a hierarchical clustering on the principal components. The details of the procedure implemented are presented in Soulat et al. (2018).
Soulat, J., B. Picard, S. Léger, M.-P. Ellies-Oury, and V. Monteils. 2018. Preliminary Study to Determinate the Effect of the Rearing Managements Applied during Heifers’ Whole Life on Carcass and Flank Steak Quality. Foods 7:160.
What is the meaning of your treatment arrangement in an industry point of view?
According to the demands of the market, the slaughterers are looking for certain types of carcasses. Our models allow to predict the potential of quality (conformation and fatness) of animals during their fattening. So, they could be used to improve the management from the rearing factors applied during the whole life of heifers and particularly during the fattening period. This could help the slaughterers to have an earlier vision of the cattle carcass types that will be slaughtered in their slaughterhouse. Our models predicting a potential of carcass and meat qualities of living cattle could be used as consulting tool to anticipate the demands of the market.
3. Explain your experimental design (CRD? RCB?) Was there any blocking in this study? Why did the number of animals differ hugely across treatments? Did you do a power analysis to determine the sample size before you started the experiment? If yes, please explain.
Our study was not an experimental study realized in an experimental farm. In our study, we have realized surveys to collect information on the rearing managements applied during the heifers’ whole life from different commercial farms. Then, we have collected meat samples on the heifers’ carcasses from the farms surveyed. For each animal, we had information on the rearing management applied during the whole life, on carcass and meat quality traits. The design of our study was presented in 2 articles (Soulat et al., 2018 a; b).
Soulat, J., B. Picard, S. Léger, M.-P. Ellies-Oury, and V. Monteils. 2018a. Preliminary Study to Determinate the Effect of the Rearing Managements Applied during Heifers’ Whole Life on Carcass and Flank Steak Quality. Foods 7:160.
Soulat, J., B. Picard, S. Léger, and V. Monteils. 2018b. Prediction of beef carcass and meat quality traits from factors characterising the rearing management system applied during the whole life of heifers. Meat Sci. 140:88–100. doi:10.1016/j.meatsci.2018.03.009.
The rearing management clusters were not controlled treatments but defined from the survey data. We have statistically constructed the different clusters which grouped similar rearing practices for each life period of heifers, as described in Soulat et al. (2018a). The links between the animals and the clusters were done after the clustering. Consequently, the number of animals is different for the different clusters. This choice was explained in Soulat et al. (2018a) and we have worked with the expertise of a statistician to validate our scientific process.
We did not realized a power analysis before our study because we had not realized an experimental study, but worked with different commercial farms. However, in this study, we had 96 animals with information on the rearing managements and carcass traits. Among the 96 animals, 77 had information on the meat (flank steak and rib) traits. We had on average 12 animals per farm and we worked on the individual data of each animal.
4. Authors need to provide P values for each comparison you make in the result section.
We think that is not necessary to add all the P-values in the text because it is redundant with the information of Table 2, so we added references to the table 2 and the figure 1 in the text. However, we added in the text, p < 0. 05 or p > 0.05 when we write, “it was significant” or “it was not significant, respectively. (lines 145, 147, 148, 153, 157,163, 164, 168, 169, 171, 177, 186, 200, 206, 210, 221, 223,227, 229)
5. There are spelling and grammatic errors throughout the manuscript, some of which have been pointed out in the specific comments.
We have read carefully the article and considered all the modification proposed in the specific comments part.
Specific comments
Line 2-4: Title: You need to mention ‘sensory, rheological, and color attributes’ since you focused on those.
Initially, we included this information in the title as asked by the reviewer. Here is the new title: “Effect of the Rearing Managements Applied during Heifers’ Whole Life on Sensory, Rheological, and Color Attributes of Longissimus Muscle and on Rheological of Four Other Muscles of the Beef Rib”. However, we think this title is too long (4 lines in the article), so we proposed the following smaller one: “Effect of the Rearing Managements Applied during Heifers’ Whole Life on Quality Traits of Five Muscles of the Beef Rib.” (lines 2 – 4).
To meet the demands of the reviewer we included its proposals (meat sensory properties; meat rheological properties; color attributes) in the key words (line 24).
Line 11-12: The studied traits were the sensory, rheological and color traits of longissimus muscle (LM) and the rheological traits of four other muscles….
‘Traits’ repeated. May need to restructure
We had modified this sentence (line 12)
Coma needed after rheological
We added the coma (line 12)
Line 13: complexus, infraspinatus, longissimus, rhomboideus and serratus….
Coma needed after rhomboideus
This punctuation mistake was observed throughout the manuscript. Please correct all of them.
We added coma in the text (lines14, 97, 103, 110, 120, 126, 158, 160, 230, 251)
Line 32-33: Many studies of the literature……
Many studies showed….
We modified in the text (line 32)
Line 35: Recent studies had shown…..
Recent studies have….
We modified in the text (line 34)
Line 40: sensory, rheological and color proprieties
Wrong spelling
We modified in the text (line 40)
Line 42: four others muscles
four other muscles
We modified in the text (line 42)
Line 50-51: Farmers respect a specific design brief [19] based mainly on a grass diet (conserved and pasture).
What is specific design brief ?
In this study, we worked with an official label of meat quality. The heifers had to respect a design brief in order to be labelled. The main characteristics of this design brief are:
- Only heifers
- Animals breeding in a specific area
- Cross breeds Aubrac x Charolais
- Slaughter between 26 and 42 months
- From 18 months, prohibition of corn (all forms: silage, grain, etc.) in the diet
- Carcass traits: fatness 2 or 3, conformation E, U, and R, in the system EUROP, weight above 280 kg.
What does this sentence mean?
We modified this sentence to improve this comprehension (lines 50 – 52).
Line 61: The WLP-E and WLP-F rearing managements had the rearing management….
The WLP-E and WLP-F rearing managements had the SAME rearing management….
Line
We added the word “same” in the sentence (line 63).
Lines 89-90: Briefly, the members of the trained tasting panel had training sessions before
90 starting the sensory evaluation of the LM.
Please explain the training. How many sessions? How many hours? What were the references used?
For the training session, there were 20 sessions (between 1 and 1.5 hours per session) before starting the sensory evaluation of our samples. As presented in Soulat et al. (2018), the assessors were selected and trained before the analyses of samples in accordance with ISO 8586 (International Organization for Standardization, 2012). We added information in the text (lines 94 – 96)
Soulat, J., B. Picard, S. Léger, M.-P. Ellies-Oury, and V. Monteils. 2018. Preliminary Study to Determinate the Effect of the Rearing Managements Applied during Heifers’ Whole Life on Carcass and Flank Steak Quality. Foods 7:160.
International Organization for Standardization (ISO). ISO_8586: Sensory Analysis—General Guidelines for the Selection, Training and Monitoring of Selected Assessors and Expert Sensory Assessors; ISO: Geneva, Switzerland, 2012; pp. 1–28.
Lines 92-93: For sensory evaluation, the LM samples were cut into steaks and cooked on a double-face grill to reach an internal temperature of 55°C.
What was the serving size? A whole steak? Or cubes? If yes what size?
The samples were cut into homogeneous pieces (size 15 x 20 x 20 mm). We added this information in the text (lines 99 – 100).
Line 135: of the LM eather…
Change this.
We deleted this word in the sentence (line 143)

Reviewer 2 Report
This paper is well written and within the scope of Foods. Some comments have been included:
General comments
When discussing the effects of whole-life period (i.e. feed-types) on carcass/meat quality traits, it would be recommendable to include nutritional data (energy, protein, carbohydrate, etc.) to provide the reader with greater understanding when interpreting the results or reproducing the study. Can the authors please include this information or reference where it may be found?
Looking at Soulat et al [10], it is apparent that the experimental heifers were not slaughtered as a single group but across a 6 month period. How did the authors account for slaughter day or time/season effects when comparing the different WLP groups to avoid the design being confounded?
Specific comments
L81 – Please include the illuminant and observer settings, and the aperture size of the colorimeter. Can the authors also please respond to why chroma and hue values were not also calculated from the CIE values, to be then analyzed?
L85 – Please include the temperature at which the samples were held chilled.
L106 – Just a comment, but it was interesting to look at the SF values of raw meat. I would have thought that this would have made comparisons to existing literature difficult.
L179-180 – Please include a reference or the results that support this difference in ADG.
L206 – Where did this P value come from? Only the LM was tested in this paper, not the RA.
Author Response
Reviewer 2 :
Yes | Can be improved | Must be improved | Not applicable | |
Does the introduction provide sufficient background and include all relevant references? | (x) | ( ) | ( ) | ( ) |
Is the research design appropriate? | (x) | ( ) | ( ) | ( ) |
Are the methods adequately described? | (x) | ( ) | ( ) | ( ) |
Are the results clearly presented? | (x) | ( ) | ( ) | ( ) |
Are the conclusions supported by the results? | (x) | ( ) | ( ) | ( ) |
Comments and Suggestions for Authors
This paper is well written and within the scope of Foods. Some comments have been included:
General comments
When discussing the effects of whole-life period (i.e. feed-types) on carcass/meat quality traits, it would be recommendable to include nutritional data (energy, protein, carbohydrate, etc.) to provide the reader with greater understanding when interpreting the results or reproducing the study. Can the authors please include this information or reference where it may be found?
The information can be found in Soulat et al. (2018). We had specified in the materials and methods section that this article used the experimental design described in Soulat et al. (2018). This new article is a complement of the published article Soulat et al. (2018). In the previous article, we presented the nutritional data collected during the surveys.
We added also in the results and discussion section, references to Soulat et al. (2018) (lines 150, 172, 179, 189, 199).
Soulat, J., B. Picard, S. Léger, M.-P. Ellies-Oury, and V. Monteils. 2018. Preliminary Study to Determinate the Effect of the Rearing Managements Applied during Heifers’ Whole Life on Carcass and Flank Steak Quality. Foods 7:160.
Looking at Soulat et al [10], it is apparent that the experimental heifers were not slaughtered as a single group but across a 6 month period. How did the authors account for slaughter day or time/season effects when comparing the different WLP groups to avoid the design being confounded?
It is true; the heifers were slaughtered across a 5 months period. For each farm, the heifers were not slaughtered at the same period although they received the same fattening diet, regardless of the season. We had not considered the time/season effect in our analyses, however, in the descriptive analysis of our data we looked at the effect of the season on the carcass traits and did not observed an effect. Consequently, we considered that the season had also no effect on the meat quality traits when the heifers received the same rearing management during their whole life.
In addition we consider that the large period of slaughter allowed to increase the diversity in the data set which allowed to define several rearing managements. Data obtained with a shorter period of slaughter would have been less genericity than those in this study.
Specific comments
L81 – Please include the illuminant and observer settings, and the aperture size of the colorimeter. Can the authors also please respond to why chroma and hue values were not also calculated from the CIE values, to be then analyzed?
We added information on the colorimeter (lines 85-86). We calculated chroma and hue values. We added the results in Tables 1 and 2. We added also sentences in the results and discussion section (lines 216 – 218)
L85 – Please include the temperature at which the samples were held chilled.
The samples were chilled at 4°C during the ageing. We added this information (line 90).
L106 – Just a comment, but it was interesting to look at the SF values of raw meat. I would have thought that this would have made comparisons to existing literature difficult.
L179-180 – Please include a reference or the results that support this difference in ADG.
We added a reference (line 189).
L206 – Where did this P value come from? Only the LM was tested in this paper, not the RA.
This P value of overall acceptability was calculated from the data of RA muscle published in Soulat et al. (2018). We wanted to compare the results obtained for the overall acceptability of RA and LM. However, as this leads to confusion, we chose to delate this text information
Soulat, J., B. Picard, S. Léger, M.-P. Ellies-Oury, and V. Monteils. 2018. Preliminary Study to Determinate the Effect of the Rearing Managements Applied during Heifers’ Whole Life on Carcass and Flank Steak Quality. Foods 7:160.

Round 2
Reviewer 1 Report
Table 2: Typical flavor: the value 6.52 and 7.0 are shown different (p= 0.01). However, the SEM is 0.45.
With an SEM of 0.45, the 6.52 and 7 can not be significantly different.
The same applies for initial juiciness and overall acceptability.
Please reconsider the statistical analysis.
Author Response
Yes | Can be improved | Must be improved | Not applicable | |
Does the introduction provide sufficient background and include all relevant references? | ( ) | (x) | ( ) | ( ) |
Is the research design appropriate? | (x) | ( ) | ( ) | ( ) |
Are the methods adequately described? | (x) | ( ) | ( ) | ( ) |
Are the results clearly presented? | ( ) | ( ) | (x) | ( ) |
Are the conclusions supported by the results? | (x) | ( ) | ( ) | ( ) |
Comments and Suggestions for Authors
Table 2: Typical flavor: the value 6.52 and 7.0 are shown different (p= 0.01). However, the SEM is 0.45.
With an SEM of 0.45, the 6.52 and 7 can not be significantly different.
The same applies for initial juiciness and overall acceptability.
Please reconsider the statistical analysis.
We had recalculated the standard error (SE) values for meat parameters. We had found our mistake in the SE calculation of the sensory parameters. Finally, in the table 2, we added a column SE at each WLP cluster and removed the SEM column.
